# Microfluidic Platform for Examination of Effect of Chewing Xylitol Gum on Salivary pH, O$_2$, and CO$_2$

Ivana Podunavac [1],*, Stevan Hinić [2], Sanja Kojić [3], Nina Jelenčiakova [2], Vasa Radonić [1], Bojan Petrović [3] and Goran M. Stojanović [3]

1 BioSense Institute, University of Novi Sad, 21000 Novi Sad, Serbia; vasarad@biosense.rs
2 Faculty of Medicine, University of Novi Sad, 21000 Novi Sad, Serbia; 2653s16@mf.uns.ac.rs (S.H.); ninajelenciakova@yahoo.com (N.J.)
3 Faculty of Technical Sciences, University of Novi Sad, 21000 Novi Sad, Serbia; sanjakojic@uns.ac.rs (S.K.); bojan.petrovic@mf.uns.ac.rs (B.P.); sgoran@uns.ac.rs (G.M.S.)
* Correspondence: ivana.podunavac@biosense.rs

**Abstract:** Miniaturization of different measurement processes and a scaled-down approach open the possibility for rapid measurements with the small amounts of samples and reagents into a compact platform with integrated sensors and different measuring components. In this paper, we report a microfluidic approach for measurements of salivary pH, dissolved O$_2$, and CO$_2$ during chewing xylitol gum The study was done with the samples of 30 healthy volunteers who were chewing a xylitol gum, and the measurements were performed in the microfluidic (MF) chip with integrated commercial PreSens sensors. Xylitol exhibited a significant effect on the pH of saliva in terms of its initial drop, which was the most significant between the 5th and 10th minutes. The effect of xylitol on the amount of oxygen and carbon dioxide in saliva cannot be confirmed. The employed microfluidic platform was shown to be applicable and effective in the analysis of salivary biomarkers relevant to caries development.

**Keywords:** microfluidics; saliva; pH; O$_2$; CO$_2$; xylitol; PMMA; xurography

## 1. Introduction

Microfluidics is the most advanced field of modern science and technology, due to a wide range of applications in different fields, such as medicine, biology, chemistry, or environmental protection [1–3]. The manipulation with a small amount of fluid is done in a microchannel network and integrated operations such as sample pretreatment, reagent storage, cell separation, mixing, and fluid separation along with micromechanical, optical, and electronic components for analysis and detection. It has the potential to integrate complex systems (Lab-on-Chip) in a miniaturized state by controlling the flow rate, providing a dynamic environment, conducting experiments in parallel, and monitoring analytes at a cellular scale [4–6] as non-invasive sample saliva becomes an important fluid for microfluidic and sensing applications due to a lot of information that describes dental and general health. The use of salivary samples for both oral and general health biomarkers' detection gains increasing attention from scientists and clinicians since saliva presents an easily accessible and sophisticated diagnostic fluid with a great potential for integration in various microfluidic setups [7]. In recent years, these systems have even exploited microelectronics, biotechnologies, and integration into smartphones to provide portable and miniaturized analytical devices. Despite the rapid development of microfluidic systems and their applications in many disciplines, a very small number of these systems are commercially and clinically available [8–10]. Of particular importance are microfluidic systems that allow simultaneous detection of multiple biomarkers in real time.

Dental caries is a complex, multifactorial disease that affects a large proportion of the general population, regardless of age, gender, or ethnicity. It is a disease that is directly de-

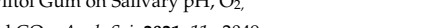

pendent on diet, microorganisms of the oral cavity, and defense mechanisms that contribute to oral homeostasis. Saliva pH is an important factor in caries' prevention due to its role in the maintenance of saturation with regard to hydroxyapatite. This protective effect of saliva or intrinsic remineralization potential strongly depends on the chemical composition of saliva, since, at physiological pH, saliva is supersaturated with phosphoprotein stabilized calcium and phosphate ions in the form of $Ca^{2+}$ and $PO_4^{3-}$ [11]. When the saliva pH or the plaque pH is below a 'critical value' of about 5.5, the saliva or plaque becomes unsaturated and demineralization processes become predominant. Changes in salivary pH are closely related to the caries' resistance properties of saliva and are, therefore, used to evaluate the intensity of tooth decay. It has been reported that the normal pH of saliva is between 6 and 7, suggesting that it is slightly acidic with the pH in the salivary flow range from 5.3 to 7.8 [12].

Data regarding $O_2$ content in human saliva are extremely rare. It is a broadly accepted fact that low $O_2$ concentration yields to cell apoptosis and tissue damage. Two reports regarding $O_2$ concentration in saliva report the range between 2 and 10 μM [13]. A few earlier studies have shown that total $CO_2$ in saliva increases with flow rate [14,15]. It has been described that the content may be less than 1 mM in unstimulated saliva and may also approach 60 mM in highly stimulated salivation. Intraoral acid-base relationships and the role of gases in this equilibrium have long been suspected of contributing to dental caries' risk, but the results in this field are contradictory [16].

Numerous agents have been described in the literature as having anticariogenic properties. Xylitol [17] is a non-fermentable sugar alcohol that has the potential to reduce Streptococcus mutans' levels by interfering with bacterial metabolic cycle by a self-destroying mechanism involving dephosphorylation of xylitol-5-phosphate [13]. Anticariogenic effects of xylitol may stem from several mechanisms [18]. In the form of gum chewing, it increases the salivary flow rate and also improves the salivary remineralization potential by increasing its pH and buffering ability [14]. Xylitol has been reported to directly decrease the growth of Streptococcus mutans and may prevent caries by inhibiting plaque formation and bacterial adherence. The effect in preventing dental decay is related to the frequency of usage and the fact that it cannot be fermented by the majority of oral bacteria. The recommended dosage for the gums is to chew them three times daily for at least 5 min within 20 min after every meal since the time frame of about 5 to 20 min just after the meal is when pH values fall down rapidly.

When it comes to the etiopathogenesis of dental caries, numerous microfluidic systems have already been developed with the aim of the detection of various biomarkers important for the progression of the caries' process. This approach opens the opportunity for a complete translation of standard macro diagnostic procedures and techniques into more precise, cost-effective, and comfortable systems. It should be emphasized here that this translation is not simple, because it requires adaptation in sampling, transfer, design of analytical procedures in accordance with the requirements, and capabilities of microfluidic systems. Previous microfluidic systems for the diagnosis of caries used various sensors to detect a wide range of biomarkers of saliva, plaque, specific metabolites, and bacterial products. It has already been emphasized that the examination of the pH of both salivary and dental plaque is an indispensable parameter in the diagnosis of the risk of caries [19,20]. Despite extensive efforts directed toward the identification of one or a combination of biomarkers with good predictive potential for the prevention, early diagnosis, and prognosis of dental caries as well as for monitoring the progression of the disease, all physical and biochemical biomarkers still need validation before chairside, point-of-care devices that can be widely used in everyday clinical practice [21].

The aim of this study was to evaluate salivary pH, $O_2$, and $CO_2$ before and after chewing sugar-free (xylitol) chewing gums in healthy volunteers using a microfluidic approach. The samples of 30 healthy volunteers were observed in the MF chip using SensorPlugs optical fiber sensors. The salivary samples were collected before gum chewing

with xylitol and during gum chewing at specific times. Collated data were analyzed using parametric statistical tests and non-parametric analyses.

## 2. Materials and Methods

### 2.1. Materials

The MF chip is manufactured by bonding three layers, made in 2-mm-thick Poly(methyl methacrylate) PMMA layers. Laser ($CO_2$ Gravograph LS1000XP laser) was used for cutting the design in PMMA, and Plotter Cutter (CE6000-60 PLUS, Graphtec America, Inc., Irvine, CA, USA) with a 45° cutting blade (CB09U) and cutting mat (12″ Silhouette Cameo Cutting Mat, Sacramento, CA, USA) was used for cutting the 3M double-sided adhesive tape (3M™ GPT-020F, St. Paul, MN 55144-1000, Minneapolis, MN, USA). The layers were bonded in the cold lamination process with the uniaxial press (Carver 3895CEB, Wabash, IN, USA). For measurements of pH, $O_2$, and $CO_2$ PreSens sensors (Precision Sensing, Regensburg, Germany), namely, OXY-1-ST, pH-1 SMA HP5, and $CO_2$-1 SMA HP5 were integrated in the MF chip. PreSense sensors are photoluminescence sensors in which pH/oxygen/carbon dioxide-sensitive polymer indicators are coated on optical fibers. They are specifically optimized for small-scale culture media and physiological solutions.

### 2.2. Methods

The proposed technology for rapid fabrication of MF chips is a hybrid technology of laser micromachining and xurography. PMMA layers were cut with $CO_2$ laser, and the bonding layers were cut with Plotter Cutter in 3M double-sided adhesive tape.

In order to prevent the sample contamination, the design of the MF chip contained the chamber for sample, which enabled measurements in the closed system environment. The chamber design contained curved edges (Figure 1, middle layer) in order to prevent forming bubbles inside the chip. The proposed design and dimensions of the channels enabled integration of different sensing components in order to get a multifunctional and compact system that enabled simultaneous measurements in the sample. The proposed design presented a proof of concept for the methodology and opened a possibility for additional improvement of the system. However, for the purpose of measurements, the chip design can be simplified to a single channel and dimensions can be additionally miniaturized for more compact systems.

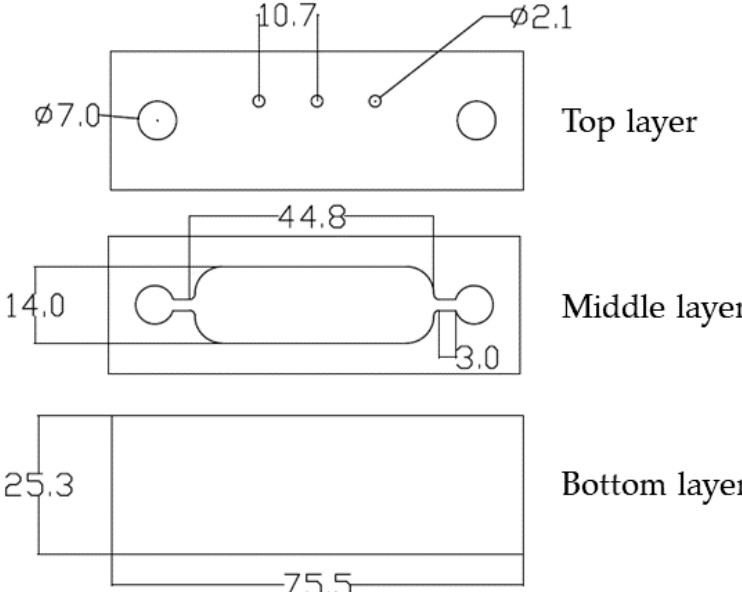

**Figure 1.** The 2D layer layouts of the microfluidic chip. The top layer contains inlet/outlet holes and three holes for the PreSens sensors. The middle layer contains chamber design, and the bottom layer closes the MF chip.

The top layer in PMMA contained an inlet and outlet customized for pipetting the sample and three holes with diameters of 2.1 mm for placing the PreSens sensors. The middle layer in PMMA contained the chamber design, which was filled with the sample, and the bottom layer that closed the MF system. The interconnecting layers, 3M double-sided adhesive tapes, were made in the same design as the middle PMMA layer, i.e., containing a chamber. Bonding of layers was rapidly achieved by the cold lamination process.

The experimental setup is presented in Figure 2. The MF chip was filled with samples and the SensorPlugs were placed in the special holes at the top layer to provide direct contact between sensors and sample. Optical fibers connected the SensorPlugs and sensors, and the whole setup was connected to a PC and software PreSens Measurement Studio 2 RC.

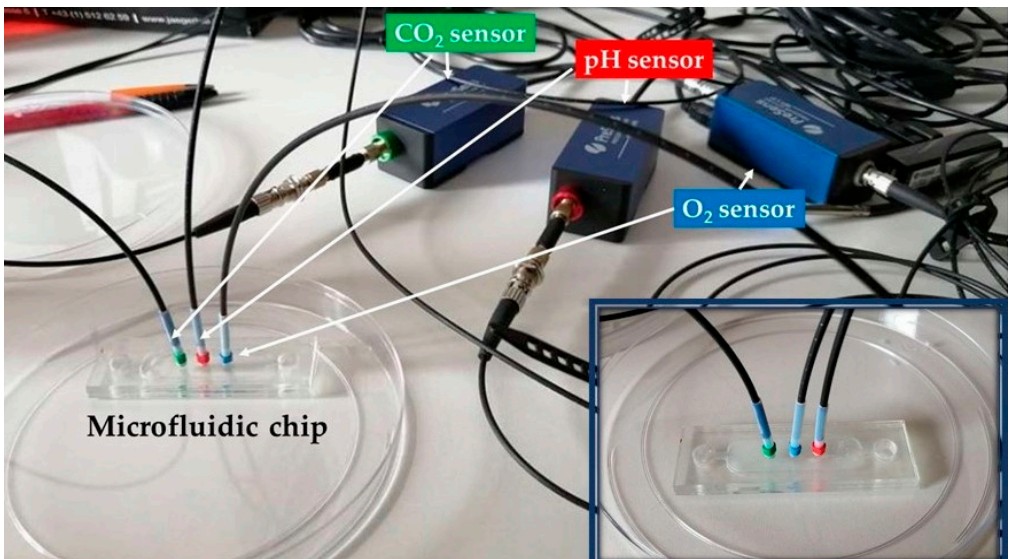

**Figure 2.** Experimental setup. Optical cables are connecting the PreSens sensors with SensorPlugs that are in direct contact with the sample in the MF chip. The PreSens sensors are connected to PC and software PreSens Measurement Studio 2 RC.

Calibration of pH and $O_2$ sensors was done according to sensors' user manual and the $CO_2$ sensor was used pre-calibrated. A benchtop pH meter was used for measurements of calibration samples with different pH and control measurements. The calibration samples were prepared in the range between 5.7 and 8.5 by titrating 0.1 M HCl in phosphate-buffered saline (PBS). The measured phase and pH values of calibration samples are presented in Table 1.

**Table 1.** Calibration parameters for pH sensor.

| pH | Phase [°] |
|---|---|
| 5.7 | 52.5810 |
| 6.3 | 48.0080 |
| 6.7 | 42.3190 |
| 8.46 | 21.7290 |

Air-saturated and oxygen-free waters were prepared for calibration of the $O_2$ sensor. Air-saturated water was made with an air pump producing bubbles in 100 mL of water during 20 min and the oxygen-free water was made by dissolving 1 g of sodium sulfite ($Na_2SO_3$) and 50 μL of cobalt nitrate ($Co(NO_3)_2$) standard solution ($\rho(Co)$ = 1000 mg/L; in nitric acid 0.5 mol/L) in 100 mL of water. The calibration values and measured phase for $O_2$ sensor are presented in Table 2.

**Table 2.** Calibration values of phase for oxygen-free and air-saturated water.

|               | Phase [°] | T [°C] |
|---------------|-----------|--------|
| Oxygen-free   | 54.88     | 27.33  |
| Air-saturated | 21.98     | 17.3   |

Thirty healthy volunteers participated in the research, 18 females and 12 males with the age range 12–56. The study participants consisted of students and staff volunteers from the Department of Dentistry, selected in line with the recommendations from the similar studies and salivary collecting protocols [22,23]. In particular, the inclusion criteria were both genders, age below 65 years, self-declaration of no basic pathologies and no pregnancy, and particularly no symptoms with respect to the salivary glands and oral mucosa. Exclusion criteria were pregnant women, participants who complain of dry mouth or dry eyes, persons with oral lesions or other contact sensitivity, subjects suffering from autoimmune diseases, and individuals with acute or chronic use of medications known to cause oral dryness. The study was approved by the Ethical Committee of Dental Clinic, Faculty of Medicine, University of Novi Sad, and all participants gave their informed consent.

The salivary samples were collected before chewing a gum with xylitol, and during chewing a gum (Orbit, Wrigley Jr. Company, Peoria, IL, USA) at specific times: after 1, 5, 10, 30 min. Salivary pH, dissolved $O_2$, and $CO_2$ were determined at baseline and after 1, 5, 10, and 30 min. The spitting method [24] was used for sample collection, and the measurements were done immediately after collecting the saliva in order to prevent $O_2$ and $CO_2$ exchange between sample and environment. The spitting method implied the accumulation of saliva in the floor of the mouth and spitting into the preweighed and graduated test tube every 30–60 s [25]. The samples were collected by spitting the saliva into the test tube, collecting the sample in a syringe, and then filling the chip with the syringe.

When it comes to statistical analyses, the data were presented in the form of mean values and standard deviation. To evaluate how far data were from normality, the Shapiro–Wilk test was used. In cases when data were with disturbed, distribution normality, the significance of the difference between the examined research groups, was tested by the Wilcoxon signed-rank test, while for the data that exhibited normal distribution one way ANOVA and paired samples T-tests were used. The level of statistical significance was set at 5% ($p < 0.05$). For all statistical calculations Jamovi software (version 0.9.2.8) was used.

## 3. Results

All participants completed the study with good compliance, and no adverse effects were reported.

The descriptive data for the entire sample (Mean, Median, SD) are presented in Table 3. As shown, the normality of the data distribution was analyzed using the Shapiro–Wilk test, and it revealed that values obtained for pH had a normal distribution, while the oxygen and carbon dioxide values deviated significantly from the normal distribution ($p < 0.05$). Consequently, all data analyses regarding salivary pH values were analyzed using parametric statistical tests, while the oxygen and carbon dioxide content were analyzed using non-parametric analyses.

Obtained data were in line with the initial drop of salivary pH values one minute after starting gum chewing, followed by additional decrease during five minutes that remained stable 10 min after baseline, with slight pH recovery 30 min from the beginning of the experiment. Paired T-test was employed for intergroup analysis and revealed that the drop of salivary pH in the period between the first and fifth minutes from the beginning of the experiment was statistically significant ($p = 0.025$, Table 4).

Since the values for oxygen and carbon dioxide content revealed non normal distribution, the Wilcoxon signed-rank test was applied for intergroup comparisons. Regarding the oxygen content, a statistically significant drop ($p = 0.025$, Table 5) was observed between

the 5th and 10th minutes, which was followed by a significant increase ($p = 0.025$, Table 4) in the period from 10 to 30 min. No significant differences between evaluated groups were observed with respect to $CO_2$.

**Table 3.** Descriptive data (mean, SD, distribution) of pH, $O_2$, and $CO_2$ measurements for the entire sample.

| | N | Mean | Median | Standard Deviation | Minimum | Maximum | Shapiro-Wilk W | Shapiro-Wilk $p$ |
|---|---|---|---|---|---|---|---|---|
| pH/bg | 30 | 6.59 | 6.5 | 0.503 | 5.67 | 7.52 | 0.976 | 0.702 |
| pH/1 | 30 | 6.52 | 6.59 | 0.522 | 5.45 | 7.78 | 0.974 | 0.644 |
| pH/5 | 30 | 6.42 | 6.54 | 0.527 | 5.13 | 7.42 | 0.959 | 0.284 |
| pH/10 | 30 | 6.42 | 6.52 | 0.573 | 4.68 | 7.23 | 0.934 | 0.063 |
| pH/30 | 30 | 6.47 | 6.52 | 0.496 | 5.46 | 7.39 | 0.971 | 0.58 |
| $O_2$/bg | 30 | 1.04 | 0.308 | 1.55 | 0.0362 | 6.37 | 0.669 | <0.001 |
| $O_2$/1 | 30 | 0.744 | 0.394 | 0.942 | 0.0447 | 3.65 | 0.645 | <0.001 |
| $O_2$/5 | 30 | 0.845 | 0.334 | 1.5 | 0.0656 | 6.92 | 0.509 | <0.001 |
| $O_2$/10 | 30 | 1.05 | 0.467 | 1.35 | 0.047 | 4.78 | 0.692 | <0.001 |
| $O_2$/30 | 30 | 0.798 | 0.333 | 1.38 | 0.0371 | 7.29 | 0.511 | <0.001 |
| $CO_2$/bg | 30 | $1.69 \times 10^6$ | $1.99 \times 10^6$ | 865,803 | 40,873 | $2.74 \times 10^6$ | 0.894 | 0.006 |
| $CO_2$/1 | 30 | $1.76 \times 10^6$ | $2.10 \times 10^6$ | 843,976 | 197,080 | $2.77 \times 10^6$ | 0.88 | 0.003 |
| $CO_2$/5 | 30 | $1.77 \times 10^6$ | $2.18 \times 10^6$ | 884,788 | 161,205 | $2.79 \times 10^6$ | 0.848 | <0.001 |
| $CO_2$/10 | 30 | $1.73 \times 10^6$ | $2.09 \times 10^6$ | 893,455 | 75,363 | $2.92 \times 10^6$ | 0.88 | 0.003 |
| $CO_2$/30 | 30 | $1.76 \times 10^6$ | $2.06 \times 10^6$ | 864,842 | 127,052 | $2.97 \times 10^6$ | 0.896 | 0.007 |

**Table 4.** Within-the-group comparison of pH at different time intervals.

| Paired Samples *t*-test | | | | | |
|---|---|---|---|---|---|
| | | | Statistic | df | $p$ |
| pH/bg | pH/1 | Student's t | 0.8206 | 29.0 | 0.419 |
| | pH/5 | Student's t | 1.7136 | 29.0 | 0.097 |
| | pH/10 | Student's t | 1.9109 | 29.0 | 0.066 |
| | pH/30 | Student's t | 1.5139 | 29.0 | 0.141 |
| pH/1 | pH/5 | Student's t | 2.3654 | 29.0 | 0.025 |
| | pH/10 | Student's t | 1.5282 | 29.0 | 0.137 |
| | pH/30 | Student's t | 1.0842 | 29.0 | 0.287 |
| pH/5 | pH/10 | Student's t | 0.0588 | 29.0 | 0.953 |
| | pH/30 | Student's t | −0.9580 | 29.0 | 0.346 |

**Table 5.** Within-the-group comparison of oxygen content at different time intervals.

| Pairwise Comparisons (Durbin-Conover) | | | |
|---|---|---|---|
| | | Statistic | $p$ |
| $O_2$/bg | $O_2$/1 | 0.995 | 0.322 |
| | $O_2$/5 | 0.332 | 0.741 |
| | $O_2$/10 | 1.990 | 0.049 |
| | $O_2$/30 | 0.166 | 0.869 |
| $O_2$/1 | $O_2$/5 | 1.326 | 0.187 |
| | $O_2$/10 | 0.995 | 0.322 |
| | $O_2$/30 | 1.161 | 0.248 |
| $O_2$/5 | $O_2$/10 | 2.321 | 0.022 |
| | $O_2$/30 | 0.166 | 0.869 |
| $O_2$/10 | $O_2$/30 | 2.155 | 0.033 |

The measured results of pH, $O_2$, and $CO_2$ for all volunteers, presented in Figures 3–5, were classified according to gender. The measured results are shown for each volunteer (divided with vertical lines) by mean value and standard deviation (SD). One way ANOVA, followed by Tukey post hoc test, revealed no statistically significant differences in obtained data in relation to gender. The pH values for male and female volunteers are presented in Figure 3a,b, respectively. The horizontal line in the graphs presents the pH = 7 and it corresponds to a neutral environment. It can be seen that 10/12 male and 15/18 female volunteers had the acidic initial pH value, and a small number of volunteers had alkaline saliva before chewing xylitol gum. In most cases, we observed that volunteers with alkaline pH and pH values higher than 6.6 had a reduced pH value after chewing gum, while volun-

teers with acidic initial pH lower than 6.6 values had an increase in salivary pH. In general, the influence of a xylitol gum to pH value is intensive during the first 10 min of chewing when the pH value starts recovering to values close to initial. In three cases (volunteers 15, 19, and 29, Figure 3) pH values dropped lower than 5.5 in at least one moment. Finally, multiple correlation analyses revealed no statistically significant correlations between the age of the participants and any of the investigated variables.

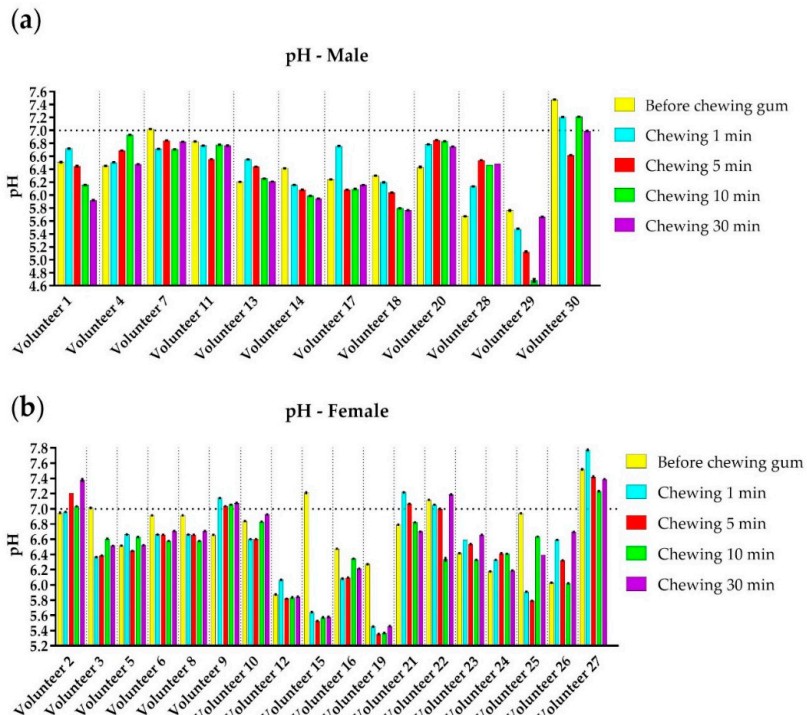

**Figure 3.** Results of pH measurements for each volunteer (**a**) male, (**b**) female. A horizontal line on graphs corresponds to neutral pH and vertical lines separate each volunteer.

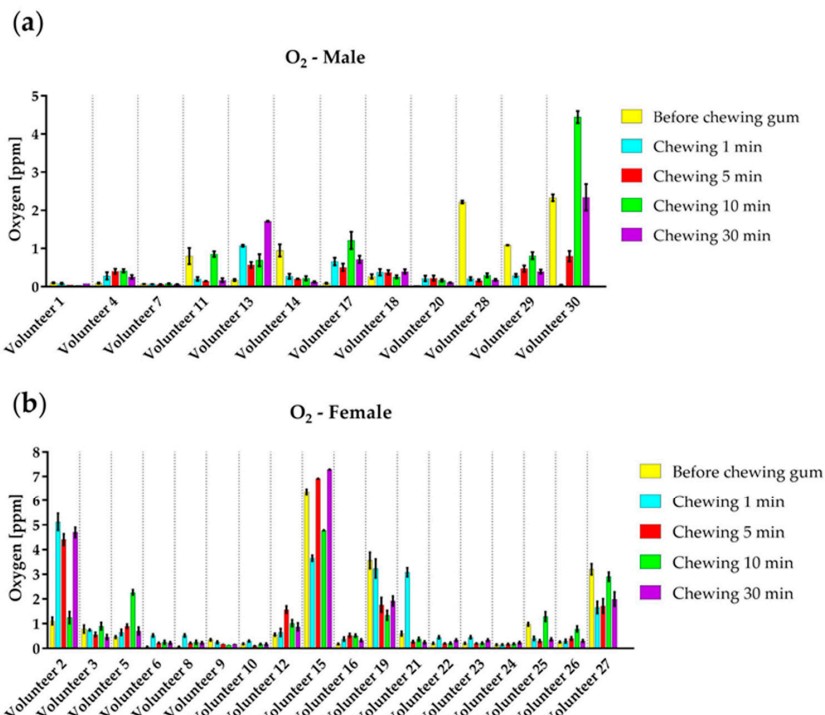

**Figure 4.** Results of $O_2$ measurements for each volunteer (**a**) male, (**b**) female.

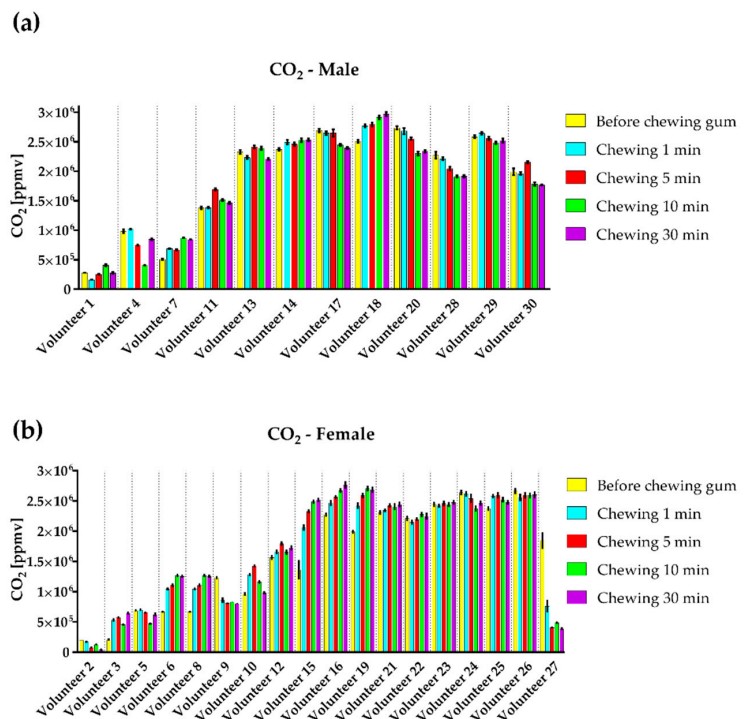

**Figure 5.** Results of $CO_2$ measurements for each volunteer (**a**) male; (**b**) female.

## 4. Discussion

This study presents the possibility of using a microfluidic platform in the analysis of salivary biomarkers of importance for the development of caries after the use of xylitol.

Saliva is a colorless, odorless, tasteless, watery liquid containing 99% water and 1% organic and inorganic substances and dissolved gases, mainly $O_2$ and $CO_2$. There is no doubt that saliva is an irreplaceable fluid, especially when it comes to caries' risk assessment. Due to its availability, non-invasive sampling, and significant diagnostic potential, saliva has been integrated into numerous microfluidic systems in the last two decades. However, the complete translation of saliva from macro to microfluidic systems is not without difficulties. The most important problems related to saliva within microfluidic systems are still related to the formation of bubbles, clogging of microfluidic channels with voluminous molecules, pronounced dimensional heterogeneity, etc. Therefore, the development of each new type of microfluidic chip imposes the need for testing, defining criteria, determining the size of the channel, standardization of mixing parameters, and fabrication and implementation of a specific analytical procedure for salivary diagnostics. On the other hand, the choice of chip and material fabrication technology is crucial to ensure biocompatibility. Therefore, the applicability of the microfluidic system in the analysis of salivary metabolites was investigated in this study.

Following preclinical studies, prior to the use of any diagnostic or therapeutic procedure, testing on healthy volunteers is a common and mandatory step, which was followed in this study as well. It is true that sample collection for saliva does not require extensive preparation, but eligible volunteers need to be adequately chosen and need to receive appropriate oral and written instructions. The fact that proper sample collection requires accurate volunteer identification, adequate sample volume, and the appropriate storage was pointed out. Finally, sample labeling and handling should be performed uniformly and precisely.

The precise routine tests that give information about oral and general health are important both for near-the-chair and self-use diagnostics. Consequently, microfluidic technology with integrated sensing components found an important field of application in healthcare, diagnostics, and theranostics. The main technological challenges in the field

are integration of MF channels with sensing components into a portable multifunctional platform, which can be easily handled by a non-expert. On the other hand, many test applications use additional reagents and chemicals. For that reason, the important topic in the field of portable system development is the integration of storage for reagents and chemicals in the MF platforms.

Besides standard Polydimethylsiloxane (PDMS) fabrication technology for MF chip fabrication [26], the other proposed technologies combine different materials and bonding solutions. Some of the commonly used technologies are realizing MF chips in glass [27], PMMA [28], low-temperature, co-fired ceramics (LTCC) [29], 3D printing [30], paper [31] etc. In addition, different hybrid fabrication technologies have been developed by combining different materials and technologies [32,33]. The technology described in this paper uses transparent, biocompatible polymer PMMA, which showed ability for rapid and precise laser manufacturing. The proposed bonding method by double-sided adhesive tape can be done in a few minutes, and the final structure did not show any leakage problems. A design of MF chip with a chamber enables integration of different electrical and optical components and, besides PreSens sensors, the proposed design has a potential for integration of additional sensors and biosensors for real-time monitoring of important saliva parameters and processes.

Xylitol is a sugar substitute that is used as a natural sweetener in sugar-free chewing gums. There is a lot of evidence supporting the use of chewing gum with xylitol in caries' prevention via different mechanisms such as increasing the salivary flow, increasing salivary pH, and enhancement of the enamel surface remineralization [8,9]. Hegde and Thakkar [10] reported a significant increase in salivary pH and flow rate after use of xylitol chewing gum. There is also a report of an increase in salivary pH with xylitol chewing gum when compared to other sugar-free chewing gums [11]. In addition, xylitol can decrease acid production in plaque, resulting in a higher plaque pH, which is a less cariogenic environment [12]. It should be noted here that many of the abovementioned effects of xylitol are indisputable and proven in clinical studies. However, when it comes to the pH, salivary pH, dental plaque pH, and overall intraoral pH are very often mentioned in the same context. These are completely different parameters from the point of view of caries' pathogenesis. Thus, when it is said that xylitol contributes to an increase in the pH of dental plaque over a long period of time, its immediate effect on the pH of saliva is neglected, which, according to recent reports, may be just the opposite. Instead of raising the salivary pH, it can be lowered. This is exactly what the results of our research show. Immediately after the start of chewing gum with xylitol, the salivary pH decreases, which is most pronounced between the 5th and 10th minutes, and then the values slowly return to the initial ones. This is important from a clinical point of view, because patients are advised to use chewing gum in the first half-hour after a meal, when the most significant acid attack is already present in the mouth, and if the saliva is initially slightly acidic, salivary pH can easily drop below the critical pH value of 5.5. This finding supports the need for some additional caution when choosing caries' preventive measures and advice on consuming xylitol chewing gum in people whose salivary pH is constitutionally lower.

Similar studies, where pH of saliva was determined during chewing xylitol gum, used a commercial pH indicator [34] or a benchtop pH meter for measurements of pH [35–40]. The measurements of the amount of $O_2$ and $CO_2$ in saliva samples are rare in the literature and the proposed solutions are based on the chemical and electrochemical reactions and calculations [41]. Our solution uses optical detection and provides a lower number of needed samples for measurements, due to the small surface of PreSens sensors that are in direct contact with the sample. The proposed solution enables the realization of multifunctional MF chips with simultaneous monitoring of different parameters in saliva and uses less than 2 mL of saliva for measurements. On the other hand, the measuring system can be additionally miniaturized and measurement of the collected sample can be performed in a totally closed system to prevent sample contamination.

The roles of gases dissolved in saliva and present in dental plaque are undoubtedly important for the development of the caries' process, but their absolute values are surprisingly rarely documented in the literature. Good oxygenation is certainly important for maintaining the integrity of soft tissues and tooth surfaces, shifting the balance from anaerobic to aerobic environment. However, this is a very complex process that is difficult to quantify in clinical settings and even more difficult to simulate in laboratory investigations. Moreover, the amount of carbon dioxide dissolved in saliva is very significant, because the bicarbonate buffer is the most important salivary buffer, on which its resistance to pH changes directly depends. Also, some previous studies have reported that changes in the amount of carbon dioxide in saliva may be associated with an increased risk of caries. The effect of xylitol on the content of oxygen and carbon dioxide in saliva could not be confirmed in the present study. Additional research is needed to further elucidate the dynamics of changes in the oxygen and carbon dioxide content dissolved in the saliva relative to other protective or demineralizing factors. The results of the amount of oxygen in this study showed that the range from the minimum and maximum measured values, i.e., 0.0362 ppm and 7.29 ppm, respectively, corresponds to the range of molarity 2.26–0.45 mM. From Figure 4 it can be seen that the high values of the $O_2$ amount are the case in the small number of volunteers; so, we can conclude that the range of oxygen corresponds to the values from the literature. Inomata and his coworkers reported that the values of $CO_2$ in saliva were similar to those in venous blood and also confirmed the strong relationship between the stimulation, duration of the stimulation, and $CO_2$ content [41]. The amount of dissolved $O_2$ and $CO_2$ did not show any regularity, due to the intensive exchange of gases with the environment during chewing a gum. The complexity of the experimental procedures, like breathing during chewing a gum or spitting the sample into glass, and then transferring from the collecting syringe to MF chip is enhancing the gas exchange between the sample and environment and it is a possible reason for no regularity in the measured results for dissolved gases. On the other hand, the results are valuable in a way that the ranges of values for dissolved $O_2$ and $CO_2$ are rare in the literature.

## 5. Conclusions

In this paper, a microfluidic approach for measurements of salivary pH, dissolved $O_2$, and $CO_2$ during chewing xylitol gum was done with the samples of 30 healthy volunteers. Within the limitations of the conducted research, it was concluded that xylitol has a significant effect on the pH of saliva in terms of its initial drop, which is the most significant between the 5th and 10th minutes after the start of chewing gum. In addition, the effect of xylitol on the amount of oxygen and carbon dioxide in saliva could not be confirmed and detailed research has to be performed to determine the effect of xylitol on their concentrations, which will be the topic of our further research. Finally, the employed microfluidic platform was shown to be applicable and effective in the analysis of salivary biomarkers relevant to caries' development.

**Author Contributions:** Conceptualization, I.P., B.P., and V.R.; methodology, I.P., B.P., and S.K.; software. I.P., V.R., S.K., and G.M.S.; validation, I.P., S.H., N.J., and S.K.; formal analysis, I.P., B.P., and S.H.; investigation, I.P., S.H., N.J., and S.K.; resources, V.R. and G.M.S.; data curation, I.P. and S.K.; writing—original draft preparation, I.P., S.H., N.J., B.P., S.K., G.M.S., and V.R.; writing—review and editing, I.P., S.H., N.J., B.P., S.K., G.M.S., and V.R.; visualization, I.P.; supervision, V.R. and G.M.S.; project administration, S.K.; funding acquisition, G.M.S. All authors have read and agreed to the published version of the manuscript.

**Funding:** This research has received funding from the European Union's Horizon 2020 research and innovation program under the Marie Sklodowska-Curie grant agreement No. 872370.

**Institutional Review Board Statement:** The study was conducted according to the guidelines of the Declaration of Helsinki and approved by the Ethics Committee of Dentistry Clinic of Vojvodina, Serbia (protocol code 01-18/15-2020 on 19 November 2020).

**Informed Consent Statement:** Informed consent was obtained from all subjects involved in the study.

**Data Availability Statement:** Data is contained within the article.

**Acknowledgments:** The authors would like to thank PreSens company for providing the equipment for Presens Sensor Plugs Competition 2020. In addition, the authors are grateful to volunteers who participated in the research.

**Conflicts of Interest:** The authors declare no conflict of interest. The funders had no role in the design of the study; in the collection, analyses, or interpretation of data; in the writing of the manuscript; or in the decision to publish the results.

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
