# Peer review of "Microfluidic Platform for Examination of Effect of Chewing Xylitol Gum on Salivary pH, O2, and CO2"

_applsci, doi:10.3390/app11052049_

Round 1

Reviewer 1 Report

In this draft paper, A microfluidic platform was used for the detection of pH O2 and CO2 in the salivary. Although the paper is thoroughly written, a few important control experiments are missing, the experimental results are not conceiving. A major revision is needed before it can be accepted for publication. 

  1. Conventionally, how the pH O2 and CO2 were measured clinically? Why the proposed microfluidic system is better. Authors also need to compare their work with recent similar publish works if there are any.
  2. Authors need to compare their system with at least one reference method, in order to confirm the accuracy of their microfluidic system.  
  3. I would like to know how many seconds (roughly) does it take from getting the sample to the sample liquid delivered to the sensing area in the microchamber. Since the O2 and CO2 exchange with the environment can be very fast. There is a need to investigate the pH, O2, and CO2 concentration change along with the time course using some standard samples. 
  4. In addition, the effect microfluidic system on the pH, CO2, and O2 is not clear, the easy way to evaluate it is to test these values immediately before and after flow through the microfluidic system (syringe/other containers, tubing, and PMMA device)

Author Response

Dear Sir/Madam,

We appreciate the careful reviewing of our manuscript and we gratefully acknowledge the reviewers and editor for their useful comments. We have modified the manuscript to answer the questions raised by the reviewers. Our answers, some additional comments and explanations are given in the text below and the changes are made in the resubmitted manuscript.

We thank you for the opportunity to improve our paper, including your comments and suggestions.

Sincerely,

The authors

Answers to the Reviewer’s Comments

Comment 1:

In this draft paper, A microfluidic platform was used for the detection of pH O2 and CO2 in the salivary. Although the paper is thoroughly written, a few important control experiments are missing, the experimental results are not conceiving. A major revision is needed before it can be accepted for publication. 

Conventionally, how the pH O2 and CO2 were measured clinically? Why the proposed microfluidic system is better. Authors also need to compare their work with recent similar publish works if there are any.

Authors’ response 1:

According to the Reviewer’s valuable remarks, the authors would like to give a more detailed explanation about the general idea and used methodology. In similar studies that analyze salivary samples, pH values of saliva are measured using standard methods based on colorimetric pH indicators or a more precise benchtop pH meter. Currently saliva-testing examination tools are used by the dental team to educate patients, assist in preventive treatment planning and properly select dental materials to initiate changes in the patient’s oral health and hygiene. Dental professionals can measure salivary pH with a pH level test strip or litmus paper. However, with the development of saliva as a diagnostic fluid, it is necessary to use much more precise methods for determining the pH of saliva, which can be useful for patients, clinicians and scientists who analyze the materials, substances and techniques used to preserve and improve oral health. Because of all this, a standard clinical protocol that would involve the use of a single sensor, especially for testing multiple biomarkers, that will meet the needs of patients, clinicians, and scientists does not yet exist. In these, early stages of salivary diagnostics, each new sensor requires validation in terms of clinical use and the ability to provide scientifically valid results.

A similar principle for sample collection was used in all studies. In the proposed study, a benchtop pH meter was used for the preparation of calibrating samples and control measurements during experiments. In that manner, the validation of the measurement results inside the microfluidic platform is confirmed with benchtop pH meter results.

Measurements of the gas amount in saliva are rare in the literature and often existing results are based on chemical and electrochemical reactions. As far as the authors know, the gold standard for measurements of oxygen and carbon dioxide in saliva does not exist. Therefore, for the best of our knowledge, we for the first time use optical sensors to determine oxygen and CO2 concentration in saliva.

The main advantage of our solution is the small amount of sample and rapid measurements. The MF chip uses less than 2 ml of saliva for measurements (and can be further decreased) which can be easily and rapidly produced by volunteers and the sample collection and transfer to the MF system can be done in 10-15 s. In addition, the narrow range of Standard Deviation results shows that the MF system disables gas exchange during measurements and enables reliable measurements in the closed system. On the other hand, the measured process can be performed in a closed system (with pumping, tubes, etc.) that will prevent sample contamination.

According to the reviewers’ suggestion, the existing paragraph in the paper is improved:

Similar studies, where pH of saliva was determined during chewing xylitol gum, used a commercial pH indicator [34], or a benchtop pH meter for measurements of pH [35–40]. The measurements of the amount of O2 and CO2 in saliva samples are rare in the literature and the proposed solutions are based on the chemical and electrochemical reactions and calculations [41]. Our solution uses optical detection and provides a lower amount of needed samples for measurements, due to the small surface of PreSens sensors that are in direct contact with the sample. The proposed solution enables the realization of multifunctional MF chips with simultaneous monitoring of different parameters in saliva and uses less than 2 ml of saliva for measurements. On the other hand, the measuring system can be additionally miniaturized and measurement of the collected sample can be performed in a totally closed system to prevent sample contamination.

Comment 2: Authors need to compare their system with at least one reference method, in order to confirm the accuracy of their microfluidic system.

Authors’ Response 2:

We would like to thank the reviewer for that valuable remark. To be more precise we would like to point out that the calibrations of the sensors were performed before the measurement. The calibration procedure is added in the resubmitted manuscript:

Calibration of pH and O22 sensors were done according to sensors’ user manual and the CO2 sensor was used pre-calibrated. A benchtop pH meter was used for measurements of calibration samples with different pH and control measurements. The calibration samples were prepared in the range between 5.7 and 8.5 by titrating 0.1 M HCl in Phosphate-buffered saline (PBS). The measured phase and pH values of calibration samples are presented in Table 1.

Table 1. Calibration parameters for pH sensor

pH

Phase [°]

5.7

52.5810

6.3

48.0080

6.7

42.3190

8.46

21.7290

An air-saturated and oxygen-free water were prepared for calibration of an O2 sensor. Air-saturated water was made with air-pump producing bubbles 100 ml of water during 20 min and the oxygen-free water was made by dissolving 1 g of sodium sulfite (Na2SO3) and 50 μl of cobalt nitrate (Co(NO3)2) standard solution (ρ(Co) = 1000 mg/l; in nitric acid 0.5 mol/l) in 100 ml of water. The calibration values and measured phase for the O2 sensor are presented in Table 2.

Table 2. Calibration values of phase for oxygen-free water and air-saturated water.

Phase [°]

T[°C]

Oxygen-free

54.88

27.33

Air-saturated

21.98

17.3

PreSens sensors are commercial sensors and the calibration procedure was done according to the user manual. Additional measurements were done after calibrations to confirm the accuracy and stability of the samples. Control measurements of pH was done with the pH meter and confirmed the comparability of the measurement inside the microfluidic platform since the good agreement was obtained in comparison with the benchtop pH meter. The air-saturated sample was measured during the time to check the decreasing trend. The negligible changes were observed in a period of 15 min. The CO2 sensor was used pre-calibrated.

Comment 3: I would like to know how many seconds (roughly) does it take from getting the sample to the sample liquid delivered to the sensing area in the microchamber. Since the O2 and CO2 exchange with the environment can be very fast. There is a need to investigate the pH, O2, and CO2 concentration change along with the time course using some standard samples.

Authors’ Response 3:

Sample collection and transfer to chip were done immediately after volunteers finished spitting. The transfer was done in roughly 10-15 s (pipetting slowly to avoid turbulences in fluid). The measurements of pH, O2, and CO2 were done for 10 min and the results in Figures 2, 3, and 4 present the Mean value with Standard Deviation. It can be seen that the range of Standard Deviation is not significant which shows that the MF system enables the closed-system measurements. Although the MF system enables reliable measurements, the main impact on O2 and CO2 amount results have the mouth environment which is an open system with constant gas exchange, chemical reactions occurring in the oral cavity, and the spitting method for sample collection. Since the same procedure for inserting the sample inside the chip and inserting the reagents for calibrations (air-saturated water and oxygen-free water) were used, the exchange on the oxygen which occurs as a consequence of inserting the sample into the chip is eliminated by calibration. 

Comment 4: In addition, the effect microfluidic system on the pH, CO2, and O2 is not clear, the easy way to evaluate it is to test these values immediately before and after flow through the microfluidic system (syringe/other containers, tubing, and PMMA device)

Authors’ response 4:

In order to avoid any additional impact on the results, the MF chip was filled with the calibration samples in the same way as saliva, and the error of gas exchange was eliminated by calibration. Furthermore, the MF system does not influence the measured values of pH, O2, and CO2 due to the static and closed system that the MF chip forms. The MF chamber is filled with the sample and sensors are placed in direct contact with the sample. The sample stays in the static conditions during measurements. On the other hand, the microfluidic chip was made from biocompatible materials that do not influence the measured values. In addition, it can be mentioned that the pH values of the samples used for calibration were measured before and after passing through the microfluidic chip using pH meters, and no change in pH was observed.

Reviewer 2 Report

Dear Authors, 

Thank you for submitting your paper, it is a relevant paper.

The topic of this article is very interesting and useful, especially for the proposed methodology.

The introduction is good, and well supported by literature.

Materials and methods are clearly described.

Statics is carried out in a proper way and graphs are good.

Results support the conclusion, but could you improve and expand a little bit the conclusion?

Author Response

Dear Sir/Madam,

We appreciate the careful reviewing of our manuscript and we gratefully acknowledge the reviewers and editor for their useful comments. We have modified the manuscript to answer the questions raised by the reviewers. Our answers, some additional comments and explanations are given in the text below and the changes are made in the resubmitted manuscript.

We thank you for the opportunity to improve our paper, including your comments and suggestions.

Sincerely,

The authors

Answers to the Reviewer’s Comments

Comment 1: Thank you for submitting your paper, it is a relevant paper. The topic of this article is very interesting and useful, especially for the proposed methodology. The introduction is good, and well supported by literature. Materials and methods are clearly described. Statics is carried out in a proper way and graphs are good. Results support the conclusion, but could you improve and expand a little bit the conclusion?

Authors’ response:

We appreciate the careful reviewing of our manuscript and we gratefully acknowledge the reviewer. Some additional explanations and comments were added to the revised manuscript according to the suggestion, as well as the comments of all reviewers.

Reviewer 3 Report

This manuscript presents a device for the measurements of salivary pH, dissolved O2, and CO2. The saliva from 30 healthy people chewing a xylitol gum was measured. The concept of evaluating the relation between salivary contents and chewing xylitol gum is interesting, but there seems to be still a huge gap between the research outcome and its clinical application. Besides, the authors did not calibrate the devices or perform a correlation test with a gold standard method. This makes the accuracy of measurements doubtful. I feel the manuscript does not quite meet the impact and innovation criteria of scientific publications. My specific comments are as follows.

(1) The authors mentioned the importance and challenges of microfluidics for clinical applications. However, I can’t see how do authors address those challenges. It is even questionable whether the device with a millimeter-scale chamber can be considered as a microfluidic system.

(2) The engineering information of the device is very limited. The authors should provide the principle/concept of the design, 2D/3D layouts, and the calibration data of the device.    

(3) The authors perform clinical tests without validating the devices, which makes the result unreliable.

Author Response

Dear Sir/Madam,

We appreciate the careful reviewing of our manuscript and we gratefully acknowledge the reviewers and editor for their useful comments. We have modified the manuscript to answer the questions raised by the reviewers. Our answers, some additional comments and explanations are given in the text below and the changes are made in the resubmitted manuscript.

We thank you for the opportunity to improve our paper, including your comments and suggestions.

Sincerely,

The authors

Answers to the Reviewer’s Comments

Comment 1: This manuscript presents a device for the measurements of salivary pH, dissolved O2, and CO2. The saliva from 30 healthy people chewing xylitol gum was measured. The concept of evaluating the relation between salivary contents and chewing xylitol gum is interesting, but there seems to be still a huge gap between the research outcome and its clinical application. Besides, the authors did not calibrate the devices or perform a correlation test with a gold standard method. This makes the accuracy of measurements doubtful. I feel the manuscript does not quite meet the impact and innovation criteria of scientific publications. My specific comments are as follows.

The authors mentioned the importance and challenges of microfluidics for clinical applications. However, I can’t see how do authors address those challenges. It is even questionable whether the device with a millimeter-scale chamber can be considered as a microfluidic system. 

Authors’ Response 1:

Authors agree with the valuable reviewers’ suggestion. Considering the definition of microfluidic systems, one of the channel dimensions has to be in the micrometer range. However, the same design of channels’ networks can be realized in different fabrication technologies, where the criteria of the microfluidic system can be satisfied. The proposed solution is a proof-of-concept for the chamber design and uses small amounts of samples and can be additionally scaled down. For that reason, authors consider that the system can be treated as a microfluidic in the broader meaning. The additional paragraph is added to explain the importance and challenges of microfluidics for clinical applications.

The precise routine tests that give information about oral and general health are important both for near-the-chair and self-use diagnostics. Consequently, microfluidic technology with integrated sensing components found an important field of application in healthcare, diagnostics and theranostics. The main technological challenges in the field are integration of MF channels with sensing components into a portable multifunctional platform, that can be easily handled by non-expert. On the other hand, many test applications use additional reagents and chemicals. For that reason, the important topic in the field of portable system development is the integration of storage for reagents and chemicals in the MF platforms. 

Comment 2: The engineering information of the device is very limited. The authors should provide the principle/concept of the design, 2D/3D layouts, and the calibration data of the device.

Authors’ Response 2:

Authors are grateful for the proposed suggestion. The additional explanations and figure of the 2D layout of each layer with dimensions is provided in the revised manuscript as well as calibration data. 

In order to prevent the sample contamination, the design of the MF chip contains the chamber for sample, that enables measurements in the closed system environment. The chamber design contains curved edges (Figure 1, middle layer) in order to prevent forming bubbles inside the chip. The proposed design and dimensions of the channels enable integration of different sensing components in order to get a multifunctional and compact system that enables simultaneous measurements in the sample. The proposed design presents a proof-of-concept for the methodology and opens a possibility for additional improvement of the system. However, for the purpose of measurements the chip design can be simplified to a single channel and dimensions can be additionally miniaturized for more compact systems.  

Please see the attachment Figure 1.

 Figure 1. 2D layer layouts of the MF chip. The top layer contains inlet/outlet holes and three holes for PreSens sensors. The middle layer contains chamber design, and the bottom layer closes the MF chip. 

Calibration of pH and O2 sensors were done according to sensors’ user manual and the CO2 sensor was used pre-calibrated. A benchtop pH meter was used for measurements of calibration samples with different pH and control measurements. The calibration samples were prepared in the range between 5.7 and 8.5 by titrating 0.1 M HCl in Phosphate-buffered saline (PBS). The measured phase and pH values of calibration samples are presented in Table 1.

Table 1. Calibration parameters for pH sensor

pH

Phase [°]

5.7

52.5810

6.3

48.0080

6.7

42.3190

8.46

21.7290

An air-saturated and oxygen-free water were prepared for calibration of an O2 sensor. Air-saturated water was made with air-pump producing bubbles 100 ml of water during 20 min and the oxygen-free water was made by dissolving 1 g of sodium sulfite (Na2SO3) and 50 μl of cobalt nitrate (Co(NO3)2) standard solution (ρ(Co) = 1000 mg/l; in nitric acid 0.5 mol/l) in 100 ml of water. The calibration values and measured phase for the O2 sensor are presented in Table 2.

Table 2. Calibration values of phase for oxygen-free water and air-saturated water.

Phase [°]

T[°C]

Oxygen-free

54.88

27.33

Air-saturated

21.98

17.3

Comment 3: The authors perform clinical tests without validating the devices, which makes the result unreliable. 

Authors’ response 3:

As far as authors know, in clinical application the gold standard for measurements of parameters pH, O2 and CO2 does not exist. Similar studies for pH measurements use commercial pH indicators or a more precise benchtop pH meters. In order to validate the calibration of the pH sensor, the control measurements with a benchtop pH meter were done with the calibrating and the test samples. The control measurements showed negligible deviations, around 1%. In addition, in order to eliminate the potential deviation due to the transfer of sample and filling the chip, the calibration samples were used in the same way as saliva samples. On the other hand, the results and methods for estimation of gas amount in saliva are rare in the literature, and few old papers are using methods based on chemical and electrochemical reactions for estimation of O2 and CO2 in saliva. In our approach, the optical sensors were used for measurements and the range of the measured results correspond to the results from the literature. In addition, the values of standard deviation indicates that the system can be treated as closed during measurements due to negligible changes of gas amount in the saliva. The potential uncertainty of the results can be a consequence of the complexity of the sample collecting method and the intensive gas exchange in the open system like mouth environment is. In order to minimize the potential error, measurements were done immediately after spitting, roughly 10-15 s after collecting the sample. Additional future studies can be done examining different methods of collecting saliva.

Round 2

Reviewer 1 Report

The feedback addressed my concerns, I would like to recommend this work to publish

Reviewer 3 Report

The Authors have addressed my comments adequately. I have no further comments.